# Impact of Backpacks on Ergonomics: Biomechanical and Physiological Effects: A Narrative Review

**DOI:** 10.3390/ijerph19116737

**Published:** 2022-05-31

**Authors:** Matteo Genitrini, Francesca Dotti, Eleonora Bianca, Ada Ferri

**Affiliations:** Dipartimento di Scienza Applicata e Tecnologia (DISAT), Politecnico di Torino, 10129 Torino, Italy; matteo.genitrini@live.com (M.G.); francesca.dotti@polito.it (F.D.); ada.ferri@polito.it (A.F.)

**Keywords:** backpack, ergonomics, gait, oxygen uptake, load carriage

## Abstract

(1) Background: the effects of load carriage packs on human gait biomechanics, physiology and metabolism depend on the weight carried, the design of the pack and its interaction with the user. (2) Methods: An extensive search in the PubMed database was performed to find all the relevant articles using the following keywords: backpack, rucksack, backpack ergonomy and sports backpack; 60 articles were included. (3) Results and significance: Double pack (DP) and T-pack (TP) designs are recommended solutions for school children, compared with backpacks (BP). For soldiers and hikers, a backpack remains the best compromise. A hip belt is recommended for BPs as well as for the back of DPs. Shorter and stiffer shoulder straps combined with a higher and tighter load placement on the back provide the best combination in terms of balance, muscle activation and energy expenditure. It is, therefore, possible to determine guidelines for designing the optimal load carriage system, depending on the application. (4) Conclusions: based on the available evidence, DP and TP are advantageous in terms of posture. DP is better than conventional BPs in terms of balance and muscle activation, but has the disadvantage of limited visibility, thermal sensation and obstructed ventilation. In general, it is desirable not to exceed 40% of body mass (BM).

## 1. Introduction

Among load carriage systems, the backpack (BP) is certainly one of the most widespread in the world, with tens of million people using them every day for shorter or longer periods since early childhood [1].

In fact, in most developed countries, students start elementary school at 5–6 and finish high school at 18–19: this means that they are carrying a load 5–6 days a week for more than a decade, which has a non-negligible impact on their physical development and psychosocial well-being.

Even among adults, the BP represents a very frequently used accessory both in professional life and in sports. Regarding sports, in particular, a comfortable, functional and well-fitting BP is crucial for both performance and safety in several disciplines, such as hiking, mountaineering, ski-mountaineering, jogging, etc.: it has been found that a suboptimal BP affects lateral stability and balance [2] and, intuitively leads to excess energy expenditure and discomfort, which can disrupt the athlete’s concentration while walking on uneven grounds and promote injuries.

The military frame is a shining example of how important a properly designed pack can be: in fact, it is not uncommon for soldiers to transport loads that exceed 40% of their body mass BM [3] for several hours in a day, with peaks of 60–70% BM when carrying the full gear [4]. Such a task cannot be performed using a BP with unevenly distributed loads or inadequately padded shoulder straps.

Overall, a BP compared to the unloaded condition results in changes in gait mechanics, metabolism, muscle activity, comfort and performance depending on the load, speed and duration of the effort [5,6,7]. Such variation in relation to the absence of equipment has also been reported for other types of sports equipment.

For running shoes compared to the unshoed condition: it has recently been suggested that the most appropriate footwear is the one that induces the least adaptations (i.e., changes) compared to unshoed walking, and allows the athlete to deviate as little as possible from the so-called preferred pattern [8]. It is reasonable to assume that this could also apply to wearing BP so that the ideal condition would produce only minor objective and subjective changes compared to unencumbered conditions.

Such alterations are physiological adaptations, and their absence would denote the lack of capacity to adapt ourselves to the changes in our environment, which would compromise our well-being. However, if alternations are excessive and prolonged, they can hinder movements or, in the worst cases, provoke fractures, paralysis, paresthesia or numbness, as well as increase fatigability [9,10,11,12].

In addition, it has been reported that the totality of adaptations depends on various factors, such as anthropometry, body mass index (BMI), training level and gender [5,13,14].

Therefore, it is important to know the above changes due to wearing in order to establish guidelines for designing an optimal carriage system.

In the present study, we reviewed articles published between 1980 and 2020 to provide a clear overview of the effects of load on ergonomics, which is understood as a combination of biomechanical, physiological, muscular, psychological and performance aspects.

## 2. Materials and Methods

### 2.1. Literature Search

A comprehensive computerized search of the PubMed electronic database was conducted in May 2020 using the following keywords: rucksack, backpack, backpack ergonomics, sport backpack.

### 2.2. Analyzed Topics

The reviewed articles included both female and male participants belonging to three main groups: schoolchildren, adult hikers/athletes and military personnel. The different typologies of packs are illustrated in Figure 1.

The reviewed articles analyzed the effects of different load carrying systems on (1) biomechanical, (2) physiological, (3) muscle activity, (4) comfort and (5) performance parameters.

### 2.3. Exclusion Criteria

The selection criteria are given:Studies not written in English were excluded;Studies that did not refer to the load-carriage-system were excluded;Studies involving subjects older than 60 were excluded;Studies not involving healthy subjects (except for idiopathic scoliosis, which is a crucial issue when considering a BP) were excluded;Studies that did not report the load as a percentage of the subject’s body weight were excluded;Studies involving computer modeling of human-equipment interaction were excluded;Reviews were excluded.

To summarize, in Figure 2 the steps of the selection process are described.

## 3. Results

### 3.1. Biomechanics

#### 3.1.1. Kinematics

##### Head, Neck, and Shoulders

**Children:** It has been reported that wearing a BP causes significantly greater forward head tilt in obese children than in normal weight students [15] and also in nonobese prepubescent children when the load overcomes 10–15% BM with respect to the unloaded condition [1,16,17].

It has been reported that only load itself, but also its placement on the back plays a role in kinematic adaptations, as a significant increase in craniovertebral angle (i.e., a smaller difference compared to the unloaded condition) was observed when carrying a load of 15% BM placed lower on the back than when placed medium or high, suggesting that a low position represents the best load [17].

The same load was found to have significantly different effects depending on the design of the carrying system: a modified DP (i.e., with most of the load in the BP and less in the front pack (FP)) promotes a more neutral posture, with respect to a BP and a DP with the load evenly distributed on the front and back; therefore, this type of design seems to be recommended [18]. Moreover, the type of carrying (i.e., on one shoulder, asymmetrically, or two shoulders, symmetrically) was found to produce significant differences in shoulder and scapulae asymmetry when a load of 10% and 15% BM was carried on one shoulder only, compared to the unloaded condition [19,20].

**Adults:** contradictory results have been reported for the head and neck. Indeed, in some studies, it was observed the head-neck inclination increased significantly when a load of 15–40% BM was applied during walking compared to the unloaded condition [21,22,23]; in contrast, other studies reported no significant differences in head position in both static conditions and during walking due to the load [24]. As for the shoulders, a significant reduction in the range of motion in the transverse plane was found when carrying a load of 25% BM compared to the unloaded condition, during inclined walking [25].

Regarding the influence of design, a comparison between BP and FP loaded up to 10–15% BM revealed significant differences in neck posture in both cases, but in the opposite direction, compared to the unloaded condition [26]. It was found that the head angle in the sagittal plane at a load of 15–25% BM was significantly different when wearing a TP compared to BP, with the former providing a posture closer to the unloaded condition than the latter [23].

Strap length has been reported to have significant effects on head and neck posture: in particular, BPs with a weight of 15% BM and long shoulder straps result in a significant reduction in the cranial spine, both compared to shorter straps and in the unloaded condition [27].

##### Trunk ad Thorax

**Children:** Obese children were found to have a significantly higher forward tilt of the trunk compared to both the unloaded condition and to normal weight students when the load increased to 15% and 20% BM [15]. In normal weight school children, during level walking or static standing position, a significant increase in the forward tilt of the trunk was observed between 0% and 10 to 25% BM [1,15,16,17,20,28] and a significant decrease in the rotational plane of the trunk in the transverse plane [29] was observed. Nevertheless, some studies reported a significant increase in backward inclination compared to unloaded conditions [30].

It was found that wearing a BP asymmetrically resulted in significant differences in trunk lateral flexion (i.e., in the frontal plane): a load of 10% and 15% BM resulted in a significant tilt on the unloaded side compared to the unloaded condition [19,20]. Moreover, a load of 15% BM did not produce significant differences in trunk forward lean depending on high, medium, or low placement on the back [1].

**Adults:** the forward tilt of the trunk increased significantly when the load was increased from the unloaded condition to 40% BM during the stance phase in both level and inclined walking [21,22,23,24,25]. In the latter, adding a load of 25% BM was found to change the coordination pattern between the shoulders and pelvis [25]. A significant decrease in the trunk rotation in the transverse plane was observed with loads of 40% BM with respect to the unloaded condition [31].

In addition, a significant interaction between walking speed and load has been found, producing greater differences between loaded and unloaded conditions in the trunk and thoracic kinematics as speed increased [25,31].

Regarding the design, a significant decrease in thoracic rotation in the transverse plane was observed only for the thorax at a load of 40% BM compared to the unloaded condition [31]. Nonetheless, a significantly higher amplitude of thoracic rotation was observed when the same load (40% BM) was carried in a BP with a hip belt compared to the no-belt condition, suggesting that the belt is beneficial; furthermore, pelvic-thoracic coordination in the transverse plane showed a more stable pattern compared to the no-belt condition [32].

In addition, it was reported that a traditional double-strap BP can induce different effects on trunk posture depending on the design of the straps: non-flexible straps caused a non-significant forward tilt of the trunk during gait with respect to the unloaded condition, while traditional straps did so when loaded up to 10% BM, suggesting that the formers as optimal [33]. It was found that the torso angle in the sagittal plane at a load of 15–25% BM was significantly different when worn with a TP compared to BP, with the former allowing a posture closer to the unloaded condition [23].

##### Spine

**Children:** Significant differences in spine length were found in school children; in particular, those carrying school BPs heavier than 10% BM presented lower values than those carrying lighter loads [34].

Further investigations on the effects of weight-bearing on lumbar lordosis are needed. In fact, some groups reported that as the weight of the BP increased, a significant decrease in the length and angle of the lumbar lordosis and the inclination of the sacrum was observed compared to the unloaded condition [34]. Conversely, other studies found no difference in lordosis angle when a load of up to 15% BM was applied [17].

**Adults:** Significant decreases in lumbar lordosis and upper thoracic kyphosis were found in adults by applying a load between 5 to 20% BM compared to the unloaded condition [35].

Moreover, it has been reported that wearing a BP as heavy as 10% BM and lumbar support significantly reduces the effect of loading on the lumbar spine compared to a BP without support, with non-significant differences compared to the unloaded condition [36].

##### Pelvis and Center of Mass—COM

**Children:** in adolescents with and without idiopathic scoliosis, a significant reduction in the range of motion of the pelvis with increasing the load up to 15–20% BM has been found in the transverse and frontal plane compared to the unloaded condition [29].

**Adults:** Similar results were reported in adults compared to the unloaded condition for loads as high as 25–40% BM in both level and inclined walking [25,31]; in addition, a significant interaction between walking speed and load was found, with grater differences produced between loaded and unloaded conditions with increasing speed [31]. Furthermore, a significant increase in pelvis anteversion was observed when the load was greater than 10% BM compared to the unloaded condition [24]. A significantly higher amplitude of pelvic rotation in the transverse plane was observed when the same load (40% BM) was carried in a BP with a hip belt compared to the no-belt condition, indicating the belt is beneficial [32].

In the last stance, it was reported that the mean height of the trajectory of the system (subject + load) COM increases with increasing load, while the shape remained similar under all loading conditions between 12.5 and 40% BM [22].

##### Lower Limbs

**Children:** In female adolescents with and without idiopathic scoliosis, a significant increase in sagittal plane hip range of motion was found with increasing loading, which was attributed to a significant increase in peak hip flexion angle during the swing and an increase in hip flexion angle during the stance phase [37,38].

In relation to the knee, conflicting results have been reported in children. Some studies reported a significant increase in peak knee flexion during the early stance when increasing BP load up to 15% BM in adolescents compared to the unloaded condition [37,38], whilst other studies found no significant differences [29].

**Adults:** Similar to children, a significant increase in sagittal plane hip range of motion was also found in adults with loads of 10–40% BM with respect to the unloaded condition [24,31,39,40], with some studies reporting an interaction between speed x and load [31].

In addition, a significant increase in hip range of motion in the sagittal plane was found in obese subjects compared to normal weight subjects carrying a load of 15% [40].

Regarding the influence of the design, FP and BP were reported to exert opposite effects, with the former promoting hip extension and the latter allowing greater hip flexion [26]. Finally, the hip angle at heel strike in the sagittal plane was found to be significantly closer to the unloaded condition with TP compared to BP with loads as high as 15–25% BM [23].

At the knee, a significant increase in flexion at foot strike and during stance was observed when a load between 20 to 40% BM was carried compared to the unloaded condition [7,39,41]. No differences were observed between obese and normal weight subjects when carrying a load of 15–20% [40].

Regarding the influence of design, the impact angle of the knee in the sagittal plane was found to be significantly different for loads of 15–25% BM when carried with a TP compared to BP [23].

At the ankle, in relation to the unloaded condition, the ranges of motion for dorsal plantar-flexion and inversion-eversion were considered and a significant increase when carrying a load of 40% BM on a slope of 15° was observed [42]. Plantar flexion during level walking was also found to increase due to load (20–40% BM) [7]. In the military population, a gradual increase of load up to 27% BM during level walking resulted in a significant increase in ankle dorsiflexion at foot strike [39], whereas contradictory results were found for dorsiflexion in the mid-stance phase, where some studies observed no significant differences due to loads of 15–30%BM [41], while others (with loads up to 27% BM) did. In addition, an overall significant increase in ankle range of motion during a complete gait cycle (i.e., one step) was observed as load increased [43].

No differences in ankle kinematics were observed when comparing loading between obese and normal weight subjects [40].

#### 3.1.2. Kinetics

##### Ground Reaction Force—GRF

**Children:** *Vertical*—Significant differences in vertical GRF (i.e., the force exerted by the ground on a body in contact with it) were found between normal weight and obese students when transporting a BP as heavy as 15–20% BM compared to the unloaded condition [15,37], with obese subjects showing higher values [15].

*Anterior-posterior—*A significant increase was found in normal weight students when the load was increased from 0 to 15–20% BM [15,37].

*Medial-lateral—*A significant increase was reported as the load is 20% BM in obese pupils [15] and 15% BM in normal weight adolescent students [37] compared to the unloaded condition.

**Adults:***Vertical—*In adults, significant increases not only in GRF impulse but also in peak value during the loading response and terminal stance as a result of loading increase by up to 35% BM were observed compared to the unloaded condition [7,23,39,42]. In other studies, the same loading produced significantly different vGRF peaks depending on pack design (TP vs. BP, with the former showing higher peaks) [23].

*Anterior-posterior—*In adults, no agreements have yet been reached in the scientific literature. Indeed, in some studies a significant increase was observed with loads of 20–30% BM compared to no load [7,42]; conversely, no differences were observed with loads of ~30–35% BM [44].

*Medial-lateral—*Conflicting results have been reported in adults: in some studies, a significant increase was observed with loads of 30–40% BM compared to no load [7], and the same was true for impulse at similar loads during inclined walking [42]; in other studies, no significant differences were found [43].

##### Joint Moments

**Children:** In adolescent students, significant increases in peak moments of hip internal and external rotation and an increase in peak moments of hip abduction and flexion were observed during stance when the load BP increased. Moreover, an increase in peak flexion moment was observed during the forward swing when the load increased up to 15% BM compared to the unloaded condition [37,38].

*Knee—*Significant increases in knee extension and valgus moments during stance were observed with a load (15% BM) compared to the unloaded condition in adolescent students [37,38].

*Ankle—*The plantarflexion moment showed a significant increase in adolescent students as the load was increased up to 15% compared to the unloaded condition [37].

**Adults:** *Hip*—In adult military personnel, significant increases in hip extension moment were observed in late stance between no load, 15% and 30% BM [41]; significant differences in frontal and sagittal hip moments were observed when carrying loads of 15% and 25% BM with a TP compared to BP, with the former yielding higher values [23].

*Knee—*Significant increases were observed in maximum knee flexion, maximum and mean valgus moment, and mean extension moment when carrying loads between 15 and 40% BM compared to the unloaded conditions [23,41,42]. The adduction moment of the knee was found to be significantly higher in loaded conditions (20% BM) than in unloaded conditions. No significant differences were found in knee moments with FP compared to BP [45], but they were significantly lower in the sagittal plane when loads of 15% and 25% BM were carried with a TP compared to traditional BP [23].

*Ankle—*In adults, significant increases in maximum and mean ankle plantarflexion moments and maximum dorsiflexion moments of the ankle when loads of 15–40% BM were applied compared to unloaded conditions [41,42].

#### 3.1.3. Spatial and Temporal Parameters

##### Cadence

**Children:** in children and adolescents with and without idiopathic scoliosis, a significant decrease in cadence was observed when the load increased by up to 15% BM with respect to the unloaded condition [37,38,46].

**Adults:** conflicting results have been reported, as some studies observed no significant effects of loading (up to 60% BM) over walking cadence compared to the unloaded condition [43,47,48], whereas other studies found significant decreases [7,49] or increases due to loads of 20–47% BM [31].

##### Stride Length

**Children:** No significant effects on stride length were found for loads up to 15% BM in children compared to the unloaded condition [38].

**Adults:** In adults, contradictory results have been reported for several load entities, as -in several studies [7,43,48] no differences were found at a load of 30–60% BM compared to 0% load, while in other studies [31] a significant decrease was observed for a load of 40% BM. Therefore, the impact of load carriage on stride length remains an open question.

##### Gait Speed

**Children**: in children and adolescents with and without idiopathic scoliosis, a significant decrease in gait speed was when the load gradually increased from 0% to 15–20% BM [29,37,38,46].

**Adults:** In the adult population, the conclusion differs significantly. Compared to the unloaded condition, some studies showed no significant effects of load on walking speed with loads up to 20% BM [47], whereas a significant decrease was observed with loads up to 40% BM in other studies [7]. However, a significant increase was found at loads of 20–35% BM load [44,50]. In all studies, loads ranged from 20–40% BM, with no clear trend in the effects as load increased or decreased. These contradictory results suggest that the influence of load and boundary conditions (type of soil, temperature, trial distance and duration) on gait speed needs further investigation.

##### Duration of Stance

**Children:** A significant increase in stance duration was observed with BP weighting 15–20% BM in both obese and normal weight children compared to the unloaded condition [15,46].

*Double support duration—*a significant increase with respect to the unloaded condition was found as the BP weight was raised to 15–20% BM in both obese and normal weight children [15,37,38,46]; in the case of obesity, a significant increase was observed compared to normal weight subjects [15].

*Single support duration—*In obese children, a significant increase was observed compared to normal weight students [15]; in non-obese subjects, a significant decrease in the duration of single support was observed for loads up to 15% BM in schoolgirls compared to the no-load scenario [37,38].

**Adults:** a significant increase in stance duration with an increasing load between 0 to 46% BM [7,49,51] was reported in adults, while the opposite result was found in inclined walking [42]. Other studies found no significant differences with loads between 5 to 15% BM compared to 0% [24].

*Double support duration—*Conflicting results have been reported. Some investigations found a significant increase in double support duration with an increasing load between 0 to 46%BM [7,49], whereas other studies found no significant differences between loaded and unloaded conditions [24,43].

*Single support duration—*a significant decrease was reported when the load was increased from 0% to 27 and 46% BM [49], while no significant differences were found for loads between 0 to 30% BM [24,43].

### 3.2. Physiology

The most studied energy-related parameter is oxygen uptake (VO_2_, i.e., diffusive oxygen transport in the lungs and microvasculature [40]): several studies in civilian and military settings have been conducted to understand the correlation between load carriage and oxygen demand.

For running, carrying a light BP (i.e., 5% BM) has been reported to produce a significant increase in VO_2_, energy cost and heart rate (HR) [51,52] with respect to unloaded conditions. A significant increase in VO_2_ due to higher loads (25–46% BM) was also observed during walking compared to the unloaded condition [53].

In the military population, carrying loads has been reported to be significantly more demanding in terms of maximal oxygen uptake (VO_2max_: this parameter provides information on the capacity of the organism to take up, transport and utilize oxygen, predominantly in contracting muscle mitochondria [40]), VO_2_, HR, pulmonary ventilation (VE) and caloric expenditure during the same performance as altitude increases [54].

Moreover, significant differences have been observed in VO_2max_, VE and HR during a 40-min marching trial at 6 km/h with 0%, 15% and 30% BM loads [41]. Consistent results were observed in studies at higher loads (30, 50 and 70% of lean subjects BM) during the same experiment, where a similar trend was observed in VO_2max_ and HR [4]; in particular, combining these results with previous studies [41], a quasi-direct relationship can be observed for load increases between 0 and 70% BM for VO_2max_ and HR.

In such studies, energy expenditure was found to be 41% VO_2max_ with a load of 30% BM [41], as high as the recommended working limits of 33–40% VO_2max_ [55,56], with other studies suggesting values up to 50% VO_2max_ [57].

Consistently, other studies have reported a load of 37% BM to require a VO_2max_ of less than 50% which is stable over time, while a load of ~60% BM, has been reported to require a relative work intensity that both significantly increases over time and is well above 50% VO_2max_ as a critical threshold that could lead to exhaustion if exceeded [58].

The design of the BP was found to have significant effects on VO_2_: specifically, DP was reported to be less exhausting than conventional BP with loads up to 30% BM. This effect is more pronounced in female subjects; no significant differences in respiratory exchange ratio (RER) in the same scenario [59]. In addition, VO_2_ and minute ventilation were reported to decrease significantly when the load (25% BM) was carried in a higher position on the back compared to a low and central position, suggesting a higher position is optimal [60]. Lastly, it has been reported that carrying a BP weighing 15% BM with a mono-shoulder strap system in schoolchildren results in a significant decrease in forced vital capacity and expiratory volume compared with a bilateral shoulder belt [61].

Relevant differences were found with regard to muscle hemodynamics. According to studies performed with LED (light emitting diode) and LDF (laser doppler flow) technologies [62,63], both mean muscle oxygenation and brachial arteries are negatively affected by load carriage.

In fact, when the weight of BP was increased by 11–23% BM [62], a decrease of up to 22 ± 23% in mean muscle oxygenation was observed. This finding was always accompanied by a sharp decrease in microvascular flow and perceived shoulder pain.

As far as brachial artery flow, a decrease of 43% has been observed in a 20% BM backpack [63]. One of the most impacting factors was the microvascular flow in the fingers, which decreased by 100%. This phenomenon has led to resulting subjective paresthesia at the hand after wearing the BP for 10 min.

Both studies have been conducted considering conventional backpacks.

On the other hand, blood flow reduction and nerve compression are two common effects of load carriage [63] and usually disappear a few minutes after the load is removed.

Intuitively, decreased blood flow corresponds to a decreased mean muscle oxygenation that led to an overall discomfort feeling in the user.

### 3.3. Muscle Activity

#### 3.3.1. Neck and Shoulders

**Children:** significant differences in neck muscle activity were found when carrying a load of 15% BM with differently designed packs. Specifically, electromyography (EMG) amplitude was lower with a modified DP, where most of the weight was on the back, compared to a traditional BP and a DP, where the load was evenly distributed on the front and back, suggesting that the modified DP is the optimal solution [18].

Trapezius activity has been reported to be significantly higher than in the unloaded condition when wearing both a traditional BP and a DP with loads up to 15% BM [1,18], yet no significant differences were observed between these two designs [18]. Additionally, no significant differences have been observed in the placement of the load (higher or lower on the back) [1].

**Adults:** No significant differences were found compared to the unloaded condition in adults with loads up to 15% BM when carrying a traditional BP [24].

It was reported that the trapezius and deltoid were significantly unloaded (i.e., showed less muscle activity) when the shoulder belt was elevated with respect to a lower configuration with a similar weight BM [64].

In addition, female university students were found to have significantly higher trapezius activity on the side on which the load is carried (when carried asymmetrically) in both the unloaded and double-strapped conditions and at loads of 10% BM [1].

#### 3.3.2. Back

**Children:** it was reported that erector spinae activation was significantly higher in children when the load was increased between 5 to 15% BM compared to the unloaded condition; the position of the load (higher or lower on the back) did not significantly change the activity [1].

**Adults:** Erector spinae activation has been reported to be significantly decreased in adults carrying a load between 5 to 15% BM compared to the unloaded condition [24]; however, other studies found no differences at comparable loads, namely 20% BM [30].

Regarding asymmetry, wearing a single strap BP resulted in significantly higher activity of the erector spinae on the contralateral side with a load of 10% BM compared to no-load condition; the latissimus dorsi showed no significant differences in its activations between single and double strap BP designs at a load of 10% BM compared to the 0% load condition [65].

#### 3.3.3. Lower Limbs

**Adults:** contradictory results have been reported: some studies found significant differences in increasing the load to 15–40% in the Tibialis Anterior, Medial and Lateral Gastrocnemius, Peroneus Longus, Biceps Femoris, Vastus Lateralis and Rectus Femoris [66,67]; other studies showed no significant differences due to a load, compared to the unloaded condition [30].

### 3.4. Comfort

#### 3.4.1. Neck and Shoulders

**Children:** it has been reported that the placement of the load and the load unity cause significant differences in neck and shoulder comfort in schoolchildren because the higher is the load (10% BM) the greater will be the perceived discomfort, which also depends on the higher or lower placement on the back [1].

**Adults:** significant differences in the perceived neck and shoulder comfort were found between traditional and more vertical load distribution, in college students wearing a BP, with the latter found to be more comfortable at a load of 10% BM [68].

Other studies found no difference in neck and shoulders comfort based on load placement, but only on load entity (15 to 40% BM, compared with 0%) [24,69,70]. In the last stand, it was reported that the rate of perceived exertion (RPE) was significantly lower when the BP was worn with the support of a hip belt [50].

Regarding the influence of design, no significant differences in shoulder comfort were found between FP and BP at a load of 10 to 15% BM [26], but the strap length has been reported to exert significant effects on shoulder pain: specifically, in BPs weighing 15% BM were observed to have significantly higher discomfort with longer shoulder straps compared to shorter ones [27].

#### 3.4.2. Back

**Adults:** significant differences in upper back comfort were observed when the load increased up to 15% BM compared to 0%. However, these differences did not depend on the placement of a higher or lower load, but only on the entity of the load [24]. On the other hand, some studies reported non-significant differences in upper back comfort at loads between 0–40% BM [70]. Therefore, the influence of load on upper back comfort remains unclear. As for RPE, it was reported to be significantly lower when the BP was worn with the support of a hip belt [50].

Regarding perceived comfort in the lower back, significant differences were found between traditional and more vertical load distribution, with the latter reported to be more comfortable when carrying a load of 10% BM [68]. Regarding the influence of the design, no significant differences in lower back comfort were found between FP and BP when the load is up to 10–15% BM [26].

#### 3.4.3. Abdomen and Waist

**Children:** the use of abdominal support in combination with a traditionally shaped BP was found to significantly reduce RPE in school children while carrying 10–20% BM compared to the same load without support [71].

The position of the load has been reported to produce significant differences in waist comfort, as the lower the load, the greater will be the perceived discomfort [1].

### 3.5. Performance

Significant changes in shot accuracy were observed in several studies because of carrying the load.

Marksmanship has been reported to decrease significantly after prolonged (45 min) load carrying at 40% BM compared to the unloaded condition [69]; shorter carriages also have a significant effect on performance in this sense: differences in marksmanship before and after a 3km walk-test were found to be due to both marching effort and load carriage; in particular, decreases in shooting accuracy was proportional to the load carried [72,73]. These results are consistent with those previously reported [3], suggesting that fatigue plays a major role in shooting accuracy.

In addition, it has been reported that wearing a BP of 15–40% BM with increasing load significantly decreases short-term and working memory compared to the same task in the unloaded conditions [73].

Rapid decision making, balance and RPE are also negatively affected by load carrying and balance disturbances when carrying up to 30% BM. Furthermore, the presence of the rucksack resulted in lower balance scores compared to the unloaded condition in every sensory perturbation scenario studied. Moreover, decision time was affected by the load as significant differences were found with respect to the no-BP condition [74].

The design of the BP has been reported to influence balance at up to 30% BM, as a DP, with load distributed both on the front and on the back, was found to be beneficial for balance in adults compared to traditional BPs, in different conditions of visibility and support [75].

## 4. Discussion

Adding a BP induces a backward shift in subjects’ COM, compensated by trunk forward inclination in order to keep COM vertically aligned to pelvis and nullify the backward moment induced by the application of the load; this expectation is corroborated by numerous studies conducted on both children and adults [1,16,18,20,21,22,23,24,25,28,39]. Likewise, asymmetrical carriage induces lateral flexion [19,20]. In the long term, both forward and lateral flexion contribute to worsening the interaction of the user with the carriage tool (i.e., the pack) inducing higher activation in trunk muscles (erector spinae and abdomen) [1,24,30].

In this sense, modified DP might provide the best trade-off in order to maintain a posture closer to the unloaded condition [18], as the front part would compensate for the aforementioned forward inclination.

If the use of traditional BPs cannot be avoided, in order not to trigger discomfort and greater trunk muscle activation, it is recommended to carry it using both shoulder straps, rather than asymmetrically.

To optimize the interaction with the pack, the use of a hip belt, and non-flexible, shorter shoulder straps would be beneficial in terms of both kinematics and comfort, as well as a more vertical load distribution [27,32,33,68]. Furthermore, in children, it is desirable not to exceed 10% BM considering the total load.

In fact, 10% BM is a critical threshold, as heavier loads may have consequences on spine lengthening, pelvis range of motion during carriage, and posture stability (the latter, in children affected by Idiopathic Scoliosis) [29,34,37,75]. In adults, higher loads are necessary to induce discomfort [24].

An alternative for school and urban purposes is a TP since this configuration has been reported to allow a posture in both upper and lower body closer to unloaded scenario compared to the traditional BPs [23].

Regarding walking, the effect of load carriage over gait spatial and temporal parameters remains unclear due to the divergent results, especially in studies conducted on adult subjects.

Conversely, the effects of load carriage on walking energetics clearly indicate that the more an activity demands, the lower the load necessary to produce significant differences compared to the unloaded condition will be, as light loads can induce increasing demands during running [52]. Walking with heavier loads produces an increase in VO_2max_ and HR proportional to the weight [3,41], and evidence indicates 50% VO_2max_ and 40% BM as thresholds that cannot be exceeded, in order to avoid exhaustion [58]. The energetic cost increases with altitude [54]. In this sense, a DP may be beneficial for both genders, but benefits are more evident in female subjects [75]; the back part should be placed higher on the back and carried with both shoulder straps to minimize energy expenditure [60,61].

## 5. Conclusions

From these findings, it emerges that a DP and TP may provide some advantages in terms of posture and comfort, nonetheless, it must be considered that the design of a tool strongly depends on its application: despite a DP possibly being advantageous in terms of balance, muscle activation and energy expenditure compared to traditional BPs [18,74,75], it also presents disadvantages, such as a reduction in visibility due to the encumbrance front part, and preventing a hiker or a soldier from seeing an unevenness on the ground in front of him/her [5]. Furthermore, this kind of design may induce heat discomfort [76] and hamper ventilation [77], in cases where the load is particularly burdensome. Intuitively, these factors concur to boost the fatigue and energy expenditure necessary to carry out a given task, making the tool (i.e., the DP) less usable, namely reducing the safety, efficiency and satisfaction of the user.

Therefore, a DP is indicated as an optimal solution for school children only, as the carriage time and the load will be lower, making it *suitable* (i.e., safe, efficient and satisfactory) for school purposes.

The same is true for TP design, which has been reported to significantly reduce postural deviations (namely head, trunk and hip kinematics) compared to the unloaded condition when compared to the traditional BPs [23]. Nevertheless, this solution would be hardly usable for hikers and soldiers due to both the higher entity of the loads usually carried in such frameworks and the mobility limitations. In fact, transporting a TP on uneven terrains (e.g., hiking trail) could result in hampered lower limb movements and consequent risks for the transporter owing to the difficulty of firmly securing the bags to the body as with traditional BPs and DPs.

Therefore, in the military, hiking and mountaineering frameworks it is recommended the use of a traditional BP with non-flexible and short shoulder straps and a hip belt, with the load vertically distributed and placed at the height of the thoracic region, rather than in a lower position; furthermore, to avoid exhaustion and discomfort, it is desirable not to exceed 40% BM.

In Table 1, the cited papers are listed according to topic and pack type.

## Figures and Tables

**Figure 1 ijerph-19-06737-f001:**
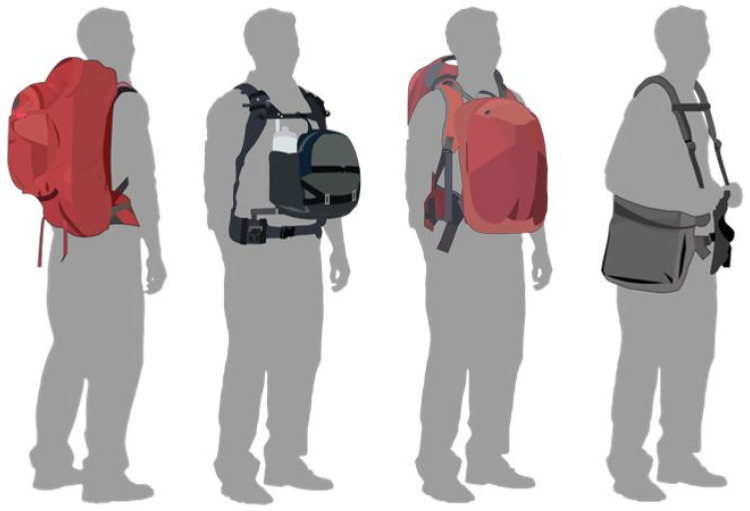
From left to right: backpack, front pack, double pack and T-pack.

**Figure 2 ijerph-19-06737-f002:**
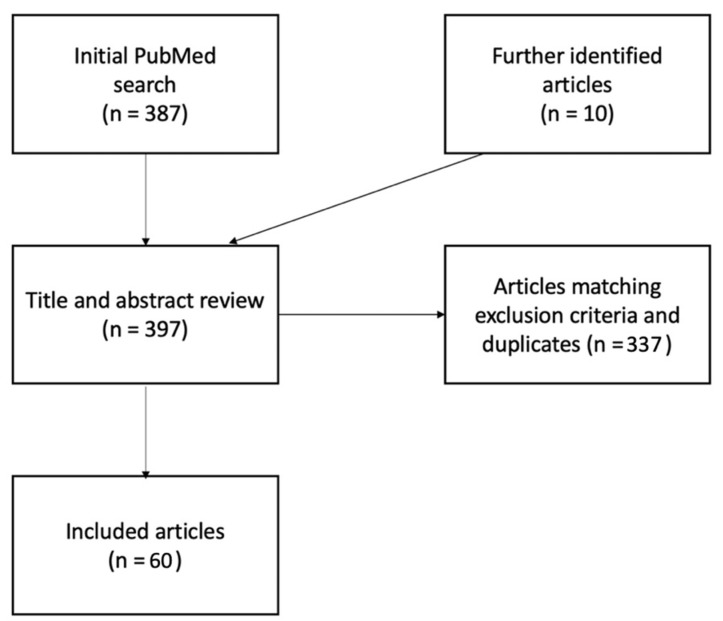
Pathway of articles identification, exclusion and inclusion.

**Table 1 ijerph-19-06737-t001:** Included articles. Abbreviations: BP = backpack, DP = double pack, FP = front pack, TP = T-pack.

Study	Year	Title	Topic	Pack Type
Abaraogu et al. [21];	(2017)	“Immediate responses to backpack carriage on postural angles in young adults: A crossover randomized self-controlled study with repeated measures”	Biomechanics	BP
Abdelraouf et al. [27];	(2016)	“Effect of backpack shoulder straps length on cervical posture and upper trapezius pressure pain threshold”	Biomechanics, comfort	BP
Ahmad and Barbosa [46];	(2019)	“The effects of backpack carriage on gait kinematics and kinetics of schoolchildren”	Biomechanics	BP
Al-Khabbaz et al. [30];	(2008)	“The effect of backpack heaviness on trunk-lower extremity muscle activities and trunk posture”	Biomechanics, muscle activity	BP
Beekley et al. [4];	(2007)	“Effects of heavy load carriage during constant-speed, simulated, road marching”	Metabolism, comfort	BP
Brackley et al. [17];	(2009)	“Effect of backpack load placement on posture and spinal curvature in prepubescent children”	Biomechanics	BP
Caron et al. [22];	(2013)	“Center of mass trajectory and orientation to ankle and knee in sagittal plane is maintained with forward lean when backpack load changes during treadmill walking”	Biomechanics	BP
Castro et al. [44];	(2015)	“The influence of gait cadence on the ground reaction forces and plantar pressures during load carriage of young adults”	Biomechanics	BP
Charteris [48];	(1998)	“Comparison of the effects of backpack loading and of walking speed on foot-floor contact patterns”	Biomechanics	BP
Chatterjee et al. [54];	(2017)	“Soldiers’ load carriage performance in high mountains: a physiological study”	Metabolism, comfort	BP
Chen and Mu [1];	(2018)	“Effects of backpack load and position on body strains in male schoolchildren while walking”	Biomechanics, muscle activity, comfort	BP
Chow et al. [37];	(2005)	“The effect of backpack load on the gait of normal adolescent girls”	Biomechanics	BP
Chow et al. [38];	(2006)	“The effect of load carriage on the gait of girls with adolescent idiopathic scoliosis and normal controls”	Biomechanics	BP
Dahl et al. [23];	(2016)	“Load distribution and postural changes in young adults when wearing a traditional backpack versus the backtpack”	Biomechanics	BP, TP
Devroey et al. [24];	(2007)	“Evaluation of the effect of backpack load and position during standing and walking using biomechanical, physiological and subjective measures”	Biomechanics, metabolism, muscle activity, comfort	BP
Drzal-Grabiec et al. [19];	(2015)	“Effects of carrying a backpack in an asymmetrical manner on the asymmetries of the trunk and parameters defining lateral flexion of the spine”	Biomechanics	BP
Epstein et al. [58];	(1988)	“External load can alter the energy cost of prolonged exercise”	Metabolism	BP
Fiolkowski et al. [26];	(2006)	“Changes in gait kinematics and posture with the use of a front pack”	Biomechanics	BP, FP
Gil-Cosano et al. [72];	(2019)	“Effect of carrying different military equipment during a fatigue test on shooting performance”	Metabolism, comfort, performance	BP
Golriz et al. [50];	(2015)	“The effect of hip belt use and load placement in a backpack on postural stability and perceived exertion: a within-subjects trial”	Biomechanics, comfort	BP
Grenier et al. [49];	(2012)	“Energy cost and mechanical work of walking during load carriage in soldiers”	Biomechanics, metabolism	BP
Hadid et al. [69];	(2017)	“Effect of load carriage on upper limb performance”	Metabolism, performance	BP
Hall et al. [45];	(2013)	“Medial knee joint loading during stair ambulation and walking while carrying loads”	Metabolism, performance	BP
Hardie et al. [45];	(2015)	“The effects of bag style on muscle activity of the trapezius, erector spinae and latissimus dorsi during walking in female university students”	Muscle activity	BP
Huang, T.P.; Kuo, A.D. et al. [51];	(2014)	“Mechanics and Energetics of Load Carriage during Human Walking”	Biomechanics	BP
In et al. [36];	(2019)	“The effects of force that pushes forward lumbar region on sagittal spinal alignment when wearing backpack”	Biomechanics	BP
B. Jacobson et al. [2];	(2003)	“Comparison of perceived comfort differences between standard and experimental load carriage system”	Comfort	BP
Jaworski, R.L.; Jensen, A. et al. [3]	(2015)	“Changes in Combat Task Performance Under Increasing Loads in Active Duty Marines”	Performance	BP
Keren et al. [53];	(1981)	“The energy cost of walking and running with and without a backpack load”	Metabolism	BP
Kim et al. [18];	(2008)	“Changes in neck muscle electromyography and forward head posture of children when carrying schoolbags”	Biomechanics, muscle activity	BP, DP
Kim et al. [63];	(2014)	“Upper Extremity Hemodynamics and Sensation with Backpack Loads”	Metabolism	BP
Kratzenstein et al. [64];	(2019)	“Height adjustments on backpack-carrying systems and muscle activity”	Muscle activity	BP
LaFiandra et al. [31];	(2003)	“How do load carriage and walking speed influence trunk coordination and stride parameters?”	Biomechanics	BP
Lee et al. [42];	(2017)	“The effect of backpack load carriage on the kinetics and kinematics of ankle and knee joints during uphill walking”	Biomechanics	BP
J. X. Li et al. [28];	(2003)	“The effect of load carriage on movement kinematics and respiratory parameters in children during walking”	Biomechanics, metabolism	BP
S. S. Li Chan et al. [75];	(2019)	“Effects of backpack and double pack loads on postural stability”	Biomechanics	BP, DP
Li, S.S.W.; Chan, O.H.T. et al. [59]	(2019)	“Gender Differences in Energy Expenditure During Walking With Backpack and Double-Pack Loads”	Biomechanics	BP
S. S. Li and Chow [35];	(2016)	“Multi-objective analysis for assessing simultaneous changes in regional spinal curvatures under backpack carriage in young adults”	Biomechanics	BP
S. S. Li, Zhen and Chow [47]	(2019)	“Changes of lumbosacral joint compression force profile when walking caused by backpack loads”	Biomechanics	BP
Lindner et al. [66];	(2012)	“The effect pf the weight of equipment on muscle activity of the lower extremity in soldiers”	Biomechanics, muscle activity	BP
Majumdar et al. [39];	(2010)	“Effects of military load carriage on kinematics of gait”	Biomechanics	BP
Majumdar et al. [43];	(2013)	“Kinetic changes in gait during low magnitude military load carriage”	Biomechanics	BP
Mallakzadeh et al. [76];	(2016)	“Analyzing the potential benefits of using a backpack with non-flexible straps”	Biomechanics, comfort	BP
Marsh et al. [71];	(2006)	“Changes in posture and perceived exertion in adolescents wearing backpacks with and without abdominal supports”	Biomechanics, comfort	BP
May et al. [74];	(2009)	“Effects of backpack load on balance and decisional processes”	Performance	BP
Mao et al. [62];	(2015)	“Shoulder Skin and Muscle Hemodynamics during Backpack Carriage”	Metabolism	BP
Negrini and Negrini [20]	(2007)	“Postural effects of symmetrical and asymmetrical loads on the spines of schoolchildren”	Biomechanics	BP
Quesada et al. [41];	(2000)	“Biomechanical and metabolic effects of varying backpack loading on simulated marching”	Biomechanics, metabolism	BP
Ramprasad et al. [16];	(2010)	“Effect of backpack weight on postural angles in preadolescent children”	Biomechanics	BP
Rosa et al. [25];	(2018)	“Inclined weight-loaded walking at different speeds: pelvis-shoulder coordination, trunk movements and cost of transport”	Biomechanics, metabolism	BP
Sahli et al. [78];	(2013)	“The effects of backpack load and carrying method on the balance of adolescent idiopathic scoliosis subjects”	Biomechanics	BP
Scheer et al. [52];	(2013)	“Running economy and energy cost of running with backpacks”	Metabolism	BP
Sharpe et al. [32];	(2008)	“Effects of a hip belt on transverse plane trunk coordination and stability during load carriage”	Biomechanics	BP
Simpson et al. [63];	(2011a)	“Backpack load affects lower limb muscle activity patterns of female hikers during prolonged load carriage”	Muscle activity	BP
Simpson et al. [70];	(2011b)	“Effect of load mass on posture, heart rate and subjective responses of recreational female hikers to prolonged load carriage”	Biomechanics, comfort	BP
Simpson et al. [7];	(2012)	“Effect of prolonged load carriage on ground reaction forces, lower limb kinematics and spatiotemporal parameters in female recreational hikers”	Biomechanics	BP
Singh and Koh [29];	(2009)	“Lower limb dynamics change for children while walking with backpack loads to modulate shock transmission to the head”	Biomechanics	BP
Smith et al. [40];	(2010)	“The Effect of Evenly Distributed Load Carrying on Lower Body Gait Dynamics for Normal Weight and Overweight Subjects”	Biomechanics	BP
Son et al. [73];	(2019)	“Effects of backpack weight on the performance of basic short-term/working memory tasks during flat-surface standing”	Performance	BP
Song et al. [15];	(2014)	“Effects of backpack weight on posture, gait patterns and ground reaction forces of male children with obesity during stair descent”	Biomechanics	BP
Stuempfle et al. [60];	(2004)	“Effect of load position on physiological and perceptual responses during load carriage with ans internal frame backpack”	Metabolism, comfort	BP
A. C. Vieira and Ribeiro [61];	(2015)	“Impact of backpack type on respiratory muscle strength abd lung function in children”	Metabolism	BP
Walicka-Cuprys et al. [34];	(2015)	“Influence of the weight of a school backpack on spinal curvature in the sagittal plane of seven-year-old children”	Biomechanics	BP

## Data Availability

Not applicable.

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
