# Peer review of "Impact of Backpacks on Ergonomics: Biomechanical and Physiological Effects: A Narrative Review"

_ijerph, 2022, doi:10.3390/ijerph19116737_

Round 1

Reviewer 1 Report

The authors made a thorough review of different biomechanical parameters affected by carriage systems in literature. The discussion and conclusion sections are overall fine, though there are lots of format/grammar mistakes the authors need to correct and be careful with. Please proofread the manuscript next time before submission.

Abstract:

“DP”, “TP”, and “BM” were used before introducing the full phrase.

Introduction:

Line 29: “Tens of million people using it” needs a reference

Line 43: In fact, it is not uncommon …

Line 50-56: The beginning of this paragraph disconnects from previous content. It would be better if the paragraph begins with alteration/deviate issues and how other sports equipment was designed to compensate for this.

Line 63: The citation numbers are not in the right order

Line 67: Missing a comma after “In the present study”

Materials and methods:

Figure 2 was referred to before Figure 1

Line 80: This paragraph seems missing a line space from the previous paragraph

Figure 1 quality needs to be improved

Results:

Line 155: This sentence is very confusing. Does the significance depend on the placement of the backpack or not?

Line 162: The capital “X” may be replaced with × or just use “interaction between A and B”

Line 204: A significant increase in pelvis anteversion has

Line 233: I would prefer a more consistent style rather than “As far as the ankle is concerned”.

Line 352: The Metabolism section doesn’t have adult/children subsections. While section 3.1.5.3 has Adults only subsections?

General:

A table that summarizes all the literature into categories would be helpful for the readers

The spaces between paragraphs seem very inconsistent throughout the manuscript

Reviewer 2 Report

Dear authors, your article seems interesting to me but difficult to finalize.
I) I found a certain emphasis in the description of obese children but the adult counterpart would be missing, it seems to me an interesting fact to include.
II) Some abbreviations are not defined, eg. line 245 GRF, the acronym is made explicit but its meaning is not specified (you could add it in the materials and methods if it is an index that you think can be useful to explain a data); line 196 and 512 COM, the acronym is not defined which honestly I did not understand what it refers to; line 543 VO2max, there is no clarification on the meaning of this index which however should explain (in an unclear way) that 50% of VO2max combined with 40% of BM are the limits of effort within which an adult (I guess) it is better not to stay for too long; finally, again in line 543 HR, another index that increases with the increase of the weight carried. The use of these indices or acronyms must be specified the first time they are used, I think that in your case it is also necessary to explain to the reader the functional meaning of using these indices.

III) Personally I think the work is well written in the results and introduction but the amount of data provided by your 64 articles analyzed could have been better organized and discussed.

IV) There are several typos such as excess spaces and small inaccuracies. I recommend a careful re-reading.

Best regards.

Reviewer 3 Report

This article represents an extensive review of the effects that using a backpack might have on health.

The authors included a considerable amount of articles, selecting most of the aspects on which such use could affect.

However, I found an incongruity. In the title of this work, physiological and psychological effects are mentioned. However, there are no papers on the latter, there is not even a section of results dedicated to psychological aspects.

Therefore, I believe that the title needs to be revised as well as the introduction and discussion.

Round 2

Reviewer 2 Report

Dear authors, the manuscript seems to have improved, however I think there are still some typos such as line 346 "Joint moments". I read the brief explanation of the VO2max parameter in paragraph 3.2, not 3.4 as you reported in the cover letter, is it correct?

At this point, re-reading the manuscript further I would say that it has improved a lot also thanks to the indication of my fellow reviewers on the title.

Best regards

Reviewer 3 Report

After major revision, the manuscript is significantly improved. 

Therefore, I suggest the paper may be accepted in its present form.